# Mathematical Analysis of a Prey–Predator System: An Adaptive Back-Stepping Control and Stochastic Approach

**Kalyan Das** [1,*], **M. N. Srinivas** [2], **V. Madhusudanan** [3] **and Sandra Pinelas** [4]

[1] Department of Mathematics, National Institute of Food Technology Entrepreneurship and Management (NIFTEM), HSIIDC Industrial Estate, Kundli 131028, Haryana, India

[2] Department of Mathematics, School of Advanced Sciences, VIT, Vellore 632014, Tamil Nadu, India; mnsrinivaselr@gmail.com

[3] Department of Mathematics, S.A. Engineering College, Chennai 600077, Tamil Nadu, India; mvmsmaths@gmail.com

[4] Departamento de Ciencias Exatas e Naturais, Academia Militar, Av. Conde Castro Guimaraes, 2720-113 Amadora, Portugal; sandra.pinelas@gmail.com

* Correspondence: daskalyan27@gmail.com

**Abstract:** In this paper, stochastic analysis of a diseased prey–predator system involving adaptive back-stepping control is studied. The system was investigated for its dynamical behaviours, such as boundedness and local stability analysis. The global stability of the system was derived using the Lyapunov function. The uniform persistence condition for the system is obtained. The proposed system was studied with adaptive back-stepping control, and it is proved that the system stabilizes to its steady state in nonlinear feedback control. The value of the system is described mostly by the environmental stochasticity in the form of Gaussian white noise. We also established some conditions for oscillations of all positive solutions of the delayed system. Numerical simulations are illustrated, and sustained our analytical findings. We concluded that controlled harvesting on the susceptible and infected prey is able to control prey infection.

**Keywords:** prey–predator system; persistence; adaptive back-stepping control; global stability; stochastic analysis

## 1. Introduction

Theoretical research and field observations have established the prevalence of various infectious diseases amongst the majority of the ecosystem population. In the ecological system, the impact of such infectious diseases is an important area of research for ecologists and mathematicians. The processes of merging ecology and epidemiology in the past few decades have been challenging and interesting. By nature, species are always dependent on other species for its food and living space. It is responsible for spreading infectious diseases and also competes against and is predated by other species. The dynamical behavior of such systems is analyzed using mathematical models that are described by differential equations. Mathematical epidemic models have gained much attention from researchers after the pioneering work of Kermack and McKendrick [1] on the SIRS (Susceptible-Infective-Removal-Susceptible) system, in which the evolution of a disease which gets transmitted upon contact is described. The influence of epidemics on predation was first studied by Anderson and May [2,3]. Hadler and Freedman [4] considered the prey–predator model in which predation is more likely on infected prey. In their model, they considered that predators only became infected from infected prey by predation. Haque and Venturino [5,6] discussed the models of diseases spreading in symbolic communities. Mukhopadhyay [7]

studied the role of harvesting and switching on the dynamics of disease propagation and/or eradication. The role of prey infection on the stability aspects of a prey–predator models with different functional responses was studied by Bairagi et al. [8]. Han et al. [9] analyzed four epidemiological models for SIS (Susceptible-Infectious-Susceptible) and SIR (Susceptible-Infectious-Recovered) diseases with standard and mass action incidents. Das [10] showed that parasite infection in predator populations stabilized prey–predator oscillations. Pal and Samanta [11] studied the dynamical behavior of a ratio-dependent prey–predator system with infection in the prey population. They proved that prey refuge had a stabilizing effect on the prey–predator interaction. Numerous examples of the prey–predator relationship with infection in the prey population have been found in various studies [12–17].

Adaptation is a fundamental characteristic of living organisms, such as in the prey–predator system and other ecological models, since they attempt to maintain physiological equilibrium in the midst of changing environmental conditions. Adaptive control is an active field in the design of control systems that helps deal with uncertainties. Back-stepping is a technique for designing stability controls for a nonlinear dynamical system, and this approach is a recursive method for stabilizing the origin of the system. The control process terminates when the final external control reaches. El-Gohary and Al-Ruzaiza [18,19] discussed the chaos and adaptive control of a three-species, continuous-time prey–predator model. Recently, Madhusudanan et al. [20] studied back-stepping control in a diseased prey–predator system. They proved that the system was globally asymptotically stable at the origin with the help of nonlinear feedback control. Numerous examples of control techniques in the prey–predator system have been found in various studies [21,22].

The rest of the paper is structured as follows. In Section 2, we formulate a mathematical model with an assumption, and the positivity and boundedness of the deterministic model is also discussed. Section 3 deals with the existence of equilibrium points with a feasible condition. In Section 4, local stability analysis of equilibrium points is discussed. Section 5 deals with global stability analysis of the interior equilibrium point $E_3(x^*, y^*, z^*)$. We discuss the condition for permanence of the system in Section 6. In Section 7, we introduce adaptive back-stepping control in the prey–predator system. In Section 8, we compute the population intensity of fluctuation due to the incorporation of noise, which leads to chaos in reality. In Section 9, we propose and analyze a delayed prey–predator system. Numerical simulation of the proposed model is presented in Section 10. Finally, the discussion is presented in Section 11 and conclusions are presented in the final section.

## 2. Mathematical Model

In this section, a continuous-time prey–predator system with susceptible, infected prey and a predator is considered. It is assumed that the susceptible prey population was developed on the basis of logistic law, and that only infected prey are predated. The disease is inherited only from the prey population, and they remain infected and do not recover.

The model becomes:

$$\frac{dx}{dt} = x\left(1 - \frac{x}{K}\right) - \frac{axy}{1+x} - h_1 x,$$

$$\frac{dy}{dt} = \frac{axy}{1+x} - byz - h_2 y, \tag{1}$$

$$\frac{dz}{dt} = fyz - dz.$$

Here, the parameters $x(t)$, $y(t)$, and $z(t)$ denote the susceptible prey, infected prey, and predator populations, respectively. The parameters $a, d, b, f, h_1$, and $h_2$ denote the rate of transmission from the susceptible to infected prey population, death rate of predators, searching efficiency of the predator, conversion-efficiency rate of the predator, and constant harvesting rate of susceptible prey and infected prey, respectively. Now, we will analyze system (1) with the following initial conditions:

$$x(0) \geq 0, y(0) \geq 0, z(0) \geq 0. \tag{2}$$

*Positiveness and Boundedness of the System*

In theoretical eco-epidemiology, boundedness of the system implies that the system is biologically well-behaved. The following theorems ensure the positivity and boundedness of the system (1):

**Theorem 1.** *All solutions of* $(x(t), y(t), z(t))$ *of system (1) with the initial condition (2) are positive for all* $t \geq 0$.

**Proof.** From (1), it is observed that

$$\frac{dx}{x} = \left[ \left( 1 - \frac{x}{K} \right) - \frac{ay}{1+x} - h_1 \right] dt = \phi_1(x, y) dt,$$

where $\phi_1(x, y) = \left( 1 - \frac{x}{K} \right) - \frac{ay}{1+x} - h_1$.

Integrating in the region $[0, t]$, we get $x(t) = x(0) \exp \left( \int \phi_1(x, y) dt \right) > 0$ for all $t$. From (1), it is observed that

$$\frac{dy}{y} = \left[ \frac{ax}{1+x} - bz - h_2 \right] dt = \phi_2(x, z) dt,$$

where $\phi_2(x, z) = \frac{ax}{1+x} - bz - h_2$.

Integrating in region $[0, t]$, we get $y(t) = y(0) \exp \left( \int \phi_2(x, z) dt \right) > 0$ for all $t$. From (1), it is observed that

$$\frac{dz}{z} = [fy - d] dt = \phi_3(y) dt,$$

where $\phi_3(y) = fy - d$.

Integrating in the region $[0, t]$, we get $z(t) = z(0) \exp \left( \int \phi_3(y) dt \right) > 0$ for all $t$. Hence, all solutions starting from interior of the first octant $(In \Re_+^3)$ remain positive for the future. $\square$

**Theorem 2.** *All the non-negative solutions of the model system (1) that initiate in* $\Re_+^3$ *are uniformly bounded.*

**Proof.** Let $x(t), y(t), z(t)$ be any solution of system (1). Since from (1) $\frac{dx}{dt} \leq x \left( 1 - \frac{x}{K} \right)$, we have $\lim\limits_{t \to \infty} \sup x(t) \leq K$. Let $\xi = x + y + \frac{b}{f} z$; therefore,

$$\frac{d\xi}{dt} = \frac{dx}{dt} + \frac{dy}{dt} + \frac{b}{f} \frac{dz}{dt}. \tag{3}$$

Substituting Equation (1) in Equation (3), we get

$$\frac{d\xi}{dt} + m\xi = x \left( (1 + m - h_1) - \frac{x}{K} \right) + (m - h_2)y + \frac{bz}{f}(m - d) \leq x \left( (1 + m - h_1) - \frac{x}{K} \right),$$

$$\frac{d\xi}{dt} + m\xi \leq \mu \text{ since } K(1 + m - h_1) = \mu,$$

where $m$ and $\mu$ are positive constants. Applying Lemma on differential inequalities [23], we obtain

$$0 \leq \xi(x, y, z) \leq (\mu/m)(1 - e^{-mt}) + (\xi(x(0), y(0), z(0))/e^{mt})$$

and, for $t \to \infty$, we have $0 \leq \xi(x, y, z) \leq (\mu/m)$. Thus, all solutions of system (1) enter into the region

$$\Gamma = \left\{ (x, y, z) \in \Re_+^3 : 0 \leq x \leq K, 0 \leq \xi \leq (\mu/m) + \epsilon, \forall \epsilon > 0 \right\}. \tag{4}$$

This completes the proof. $\square$

## 3. Equilibrium Points and Criteria for Existence

The possible steady states for system (1) and their existence conditions for each of them are as follows:

1. The vanishing equilibrium point, $E_0 = (0, 0, 0)$, always exists.
2. The disease and predator free equilibrium point, $E_1 = (\tilde{x}, 0, 0)$, where $\tilde{x} = K(1 - h_1)$, exists provided that $h_1 < 1$.
3. The predator free equilibrium point, $E_2 = (\hat{x}, \hat{y}, 0)$, where $\hat{x} = (h_2/(a - h_2))$ and $\hat{y} = [(K(a - h_2)(1 - h_1) - h_2)/((a - h_2)^2 K)]$, exists provided that $a > h_2, h_1 < 1, K(a - h_2)(1 - h_1) > h_2$.
4. The steady state, $E_3 = (x^*, y^*, z^*)$, where $y^* = d/f$ and $z^* = [((a - h_2)x^* - h_2)/(b(1 + x^*))]$, exists.

However, $x^*$ is a positive root of (5)

$$Ax^{*2} + Bx^* + C = 0, \tag{5}$$

where $A = f, B = Kfh_1 + f - Kf, C = Kfh_1 + adK - Kf$.

By Discard's rule of sign, Equation (5) has a unique positive root, if $Kh_1 + 1 < K, fh_1 + ad > f$.

## 4. Stability Analysis

In this section, we analyzed the local stability of system (1) that is examined by constructing the Jacobian matrix relating to every equilibrium point. The Jacobian matrix of the system at any point $(x, y, z)$ is given by

$$J(x, y, z) = \begin{pmatrix} 1 - \dfrac{2x}{K} - h_1 - \dfrac{ay}{(1+x)^2} & \dfrac{-ax}{(1+x)} & 0 \\ \dfrac{ay}{(1+x)^2} & \dfrac{ax}{1+x} - bz - h_2 & -by \\ 0 & fz & fy - d \end{pmatrix}.$$

1. The variational matrix for the equilibrium point at $E_0(0, 0, 0)$ is

$$J(E_0) = \begin{pmatrix} 1 - h_1 & 0 & 0 \\ 0 & -h_2 & 0 \\ 0 & 0 & -d \end{pmatrix}.$$

The eigenvalues of $J(E_0)$ are $\lambda_1 = 1 - h_1, \lambda_2 = -h_2, \lambda_3 = -d$. All the eigenvalues are negative if $h_1 > 1$. Hence, $E_0$ is locally asymptotically stable in the $x$–$y$–$z$ direction.

2. The variational matrix for the equilibrium point at $E_1(K(1 - h_1), 0, 0)$ is

$$J(E_1) = \begin{pmatrix} h_1 - 1 & \dfrac{-aK(1 - h_1)}{1 + K(1 - h_1)} & 0 \\ 0 & \dfrac{aK(1 - h_1)}{1 + K(1 - h_1)} - h_2 & 0 \\ 0 & 0 & -d \end{pmatrix}.$$

The eigenvalues of $J(E_1)$ are $\lambda_1 = h_1 - 1, \lambda_2 = \dfrac{aK(1 - h_1)}{1 + K(1 - h_1)} - h_2, \lambda_3 = -d$. All the eigenvalues are negative if $h_1 < 1$ and $aK(1 - h_1) < h_2(1 + K(1 - h_1))$. Hence, $E_1$ is asymptotically stable in the $x$–$y$–$z$ direction.

3. The variational matrix for the equilibrium point at $E_2(\hat{x}, \hat{y}, 0)$ is

$$
J(E_2) = \begin{pmatrix} 1 - \dfrac{2\hat{x}}{K} - \dfrac{a\hat{y}}{(1+\hat{x})^2} - h_1 & -h_2 & 0 \\ \dfrac{K(a - h_2)(1 - h_1) - h_2}{Ka} & 0 & -b\hat{y} \\ 0 & 0 & f\hat{y} - d \end{pmatrix}.
$$

The eigenvalues of $J(E_3)$ are negative if it satisfies $f\hat{y} < d$ and $K(1 - h_1)(1 + \hat{x})^2 < (2\hat{x}(1 + \hat{x})^2 + aK\hat{y})$. Hence, the equilibrium point $E_2$ is locally asymptotically stable in the $x$–$y$–$z$ direction.

**Theorem 3.** *The co-existent equilibrium point $E_3(x^*, y^*, z^*)$ of system (1) exists. Then, $E_3$ is locally asymptotically stable if the following conditions satisfy*

1. $ax^* < (bz^* + h_2)(1 + x^*)$ *and*
2. $K(1 - h_1)(1 + x^*)^2 < (2x^*(1 + x^*)^2 + aKy^*)$.

**Proof.** The variational matrix at the interior point $E_3(x^*, y^*, z^*)$ is

$$
J(E_3) = \begin{pmatrix} a_{11} & a_{12} & 0 \\ a_{21} & a_{22} & a_{23} \\ 0 & a_{32} & 0 \end{pmatrix},
$$

where

$$
a_{11} = 1 - \frac{2x^*}{K} - h_1 - \frac{ay^*}{(1+x^*)^2} \qquad a_{12} = \frac{-ax^*}{1 + x^*} \qquad a_{21} = \frac{ay^*}{(1+x^*)^2},
$$

$$
a_{22} = \frac{ax^*}{1 + x^*} - bz^* - h_2 \qquad a_{23} = -by^* \qquad a_{32} = fz^*.
$$

The characteristic equations of Jacobian matrix $J(E_3)$ is given by $\lambda^3 + A_1\lambda^2 + A_2\lambda + A_3 = 0$, where

$$
A_1 = -a_{11} - a_{22}, \; A_2 = a_{11}a_{22} - a_{12}a_{21} - a_{23}a_{32}, \; A_3 = a_{11}a_{23}a_{32}
$$

and

$$
A_1A_2 - A_3 = a_{11}a_{12}a_{21} + a_{12}a_{21}a_{22} + a_{22}a_{23}a_{32} - a_{11}^2a_{22} - a_{11}a_{22}^2.
$$

The sufficient conditions for $A_1A_2 - A_3 > 0$ are $a_{11} \leq 0$, $a_{22} \leq 0$, which implies

$$
ax^* < (bz^* + hz)(1 + x^*), K(1 - h_1)(1 + x^*)^2 < [2x^*(1 + x^*)^2 + aKy^*].
$$

□

## 5. Global Stability Analysis

In this section, we investigated the global stability behavior of the system (1) at the interior equilibrium $E_3(x^*, y^*, z^*)$ by using the Lyapunov stability theorem.

**Theorem 4.** *If $(yx^*/y^*) < x < x^*$ or $x^* < x < (yx^*/y^*)$, then $E_3(x^*, y^*, z^*)$ is globally asymptotically stable.*

**Proof.** Let us define

$$
V(x, y, z) = P\left(x - x^* - x^*\ln\left(\frac{x}{x^*}\right)\right) + \left(y - y^* - y^*\ln\left(\frac{y}{y^*}\right)\right) \\ + Q\left(z - z^* - z^*\ln\left(\frac{z}{z^*}\right)\right), \tag{6}
$$

where $P, Q$ are positive constants to be chosen later.

Differentiating (6) along the solution of the system (1) with respect to $t$, we get

$$
\begin{aligned}
\frac{dV}{dt} &= P\left(\frac{x - x^*}{x}\right)\frac{dx}{dt} + \left(\frac{y - y^*}{y}\right)\frac{dy}{dt} + Q\left(\frac{z - z^*}{z}\right)\frac{dz}{dt} \\
&= P\left(1 - \frac{x}{K} - \frac{ay}{1 + x} - h_1\right)(x - x^*) + (y - y^*)\left(\frac{ax}{1 + x} - bz - h_2\right) + Q(z - z^*)(fy - d) \\
&= P\left(-\frac{1}{K}(x - x^*) - a\left(\frac{y}{1 + x} - \frac{y^*}{1 + x}\right)\right)(x - x^*) \\
&\quad + (y - y^*)\left(\frac{ax}{1 + x} - \frac{ax^*}{1 + x^*} - b(z - z^*)\right) + Q(f(z - z^*)(y - y^*)).
\end{aligned}
$$

Choosing $P = 1, Q = b/f$, and then simplified to

$$
\frac{dV}{dt} = -\frac{1}{K}(x - x^*)^2 - a\frac{(yx^* - xy^*)}{(1 + x)(1 + x^*)}(x - x^*).
$$

Now,

$$
\frac{dV}{dt} < 0 \text{ if } x^* < x < \frac{yx^*}{y^*} \text{ or } \frac{yx^*}{y^*} < x < x^*.
$$

Then, $\dfrac{dV}{dt}$ is negative definite. Consequently, $V$ is a Lyapunov function with respect to all solutions in the interior of the positive octant. $\square$

## 6. Permanence of the System

In this section, our main intuition is that the long time survival of species in an ecological system. Many notions and terms are identified in the literature to discuss and analyze the long-term survival of populations. Out of such, permanence and persistence are the ones to better analyze the system. From an ecological point of view, permanence of a system means that the long-term survival of all populations of the system.

**Definition 1.** *The system (1) is said to be permanent if $\exists\ M \geq m > 0$, such that for any solution of $(x(t), y(t), z(t))$ of system (1), $(x(0), y(0), z(0)) > 0$,*

$$
m \leq \lim_{t \to \infty}(\inf(x(t))) \leq \lim_{t \to \infty}(\sup(x(t))) \leq M,
$$

$$
m \leq \lim_{t \to \infty}(\inf(y(t))) \leq \lim_{t \to \infty}(\sup(y(t))) \leq M,
$$

$$
m \leq \lim_{t \to \infty}(\inf(z(t))) \leq \lim_{t \to \infty}(\sup(z(t))) \leq M.
$$

Now, we show that system (1) is uniformly persistent. The survival of all populations of the system in the future time is nothing but persistence in the view of ecology.

In the mathematical point of view, persistence of a system means that a strictly positive solution does not have omega limit points on the boundary of the non-negative cone.

**Definition 2.** *A population is said to be uniformly persistent if there exists $\delta > 0$, independent of $x(0)$ where $x(0) > 0$, such that*

$$
\lim_{t \to \infty}(\inf(x(t))) > \delta.
$$

**Theorem 5.** *The system (1) is uniformly persistent if*

$$
f\hat{y} - d > 0 \qquad \text{holds.} \tag{7}
$$

**Proof.** We will prove this theorem by the method of Lyapunov average function.

Let the average Lyapunov function for the system (1) be $\sigma(X) = x^p y^q z^r$, where $p, q, r$ are positive constants. Clearly, $\sigma(X)$ is non-negative function defined in $D$ of $\Re_+^3$, where

$$\Re_+^3 = \{(x, y, z), x > 0, y > 0, (1 + x)^2 - ayK > 0\}.$$

Then, we have $\Psi(X) = \dfrac{\dot\sigma(X)}{\sigma(X)} = p\dfrac{\dot x}{x} + q\dfrac{\dot y}{y} + r\dfrac{\dot z}{z}$,

$$\Psi(X) = p\left(\left(1 - \frac{x}{K}\right) - \frac{ay}{1+x} - h_1\right) + q\left(\frac{ax}{1+x} - bz - h_2\right) + r(fy - d). \tag{8}$$

Furthermore, there are no periodic orbits in the interior of positive quadrant of $x$–$y$ plane. Thus, to prove the uniform persistence of the system, it is enough to show that $\Psi(X) > 0$ in $\Re_+^3$ for a suitable choice of $p, q, r > 0$ :

$$\Psi(E_0) = p(1 - h_1) - q(h_2) - rd > 0, \tag{9}$$

$$\Psi(E_1) = q\left(\frac{aK(1 - h_1)}{1 + K(1 - h_1)} - h_2\right) - rd > 0, \tag{10}$$

$$\Psi(E_2) = r(f\hat y - d) > 0. \tag{11}$$

We noticed that, by increasing the value of $p$, while $h_1 < 1$ , $p(1 - h_1) > (qh_2 + rd)$, $\Psi(E_0)$ can be made positive. Thus, the inequality (9) holds. If $qaK(1 - h_1) > (qh_2 + rd)(1 + K(1 - h_1))$, then $\Psi(E_1)$ is positive. Thus, the inequality (10) holds. If the inequality in Equation (7) holds, then (11) is satisfied.  □

## 7. Introduction of Adaptive Back-Stepping Control in a Prey–Predator System

Adaptive back-stepping method is the back-stepping control and adaptive laws. The back-stepping design is initiated with the first state equation whose state variable has the highest integration order from the control input. The second state variable is considered as the virtual control and the stabilizing function is replaced by it. This stabilizing function can stabilize the first state variables and we set the error between the virtual control and stabilizing function as $\eta$. Then, for the second state equation, we will design a new stabilizing law to replace the third state variable for the second order system, and step back to control the signal. From the steps above, we can see that the term back-stepping means that we use the latter state as a virtual control to stabilize the previous one. The Lyapunov direct method is utilized from the stabilization method for the error between virtual control and stabilizing function. The control Lyapunov function is to be used which will be a positive definite and includes the quadratic form of the errors. In this section, the system with susceptible prey, infected prey and a predator population controlled by back-stepping using a nonlinear feedback control approach is studied. We initiate the study by assuming that system (1) can be written in the suitable form by introducing nonlinear feedback control inputs $u_1, u_2, u_3$ into the system to better analyze the prey–predator interactions:

$$\frac{dx}{dt} = x\left(1 - \frac{x}{K}\right) - \frac{axy}{1+x} - h_1 x + u_1, \tag{12}$$

$$\frac{dy}{dt} = \frac{axy}{1+x} - byz - h_2 y + u_2, \tag{13}$$

$$\frac{dz}{dt} = fyz - dz + u_3, \tag{14}$$

where $u_1, u_2, u_3$ are back-stepping nonlinear feedback controllers that are the functions of state variables and will be suitable choices to make the trajectories of the whole system (12)–(14). As long as this feedback stabilizes the system (1), $\lim\limits_{t\to\infty} ||x(t)|| = 0$ converges.

**Theorem 6.** *A diseased prey–predator system (12)–(14) is globally asymptotically stable provided the following adaptive nonlinear controls*

$$u_1 = \frac{x^2}{K} - 2x + \hat{h}_1 x, \tag{15}$$

$$u_2 = \frac{ax^2}{1+x} - \frac{ax\eta_1}{1+x} + \hat{h}_2\eta_1 - \eta_1, \tag{16}$$

$$u_3 = (b\eta_1^2 - f\eta_1\eta_2 + d\eta_2 - \eta_2), \tag{17}$$

*with the errors*

$$\eta_1 = y - v_1(x,y), \tag{18}$$

$$\eta_2 = z - v_2(x,y,z). \tag{19}$$

**Proof.** Consider the parameter estimators

$$e_a = a - \hat{a}, e_b = b - \hat{b}, e_{h_1} = h_1 - \hat{h}_1, e_{h_2} = h_2 - \hat{h}_2, e_d = d - \hat{d}, e_f = f - \hat{f}. \tag{20}$$

Considering Equation (12), the Lyapunov function of $x$ is given by

$$V_1(x, e_a, e_{h_1}) = \frac{1}{2}x^2 + \frac{1}{2}e_a^2 + \frac{1}{2}e_{h_1}^2. \tag{21}$$

By using the derivative of (20),

$$\dot{e}_a = -\dot{\hat{a}}, \dot{e}_b = -\dot{\hat{b}}, \dot{e}_{h_1} = -\dot{\hat{h}}_1, \dot{e}_{h_2} = -\dot{\hat{h}}_2, \dot{e}_d = -\dot{\hat{d}}, \dot{e}_f = -\dot{\hat{f}}.$$

Differentiating (21) with respect to $t$, we have

$$\dot{V}_1 = x\left[x\left(1 - \frac{x}{K}\right) - \frac{axy}{1+x} - h_1 x + u_1\right] + e_a(-\dot{\hat{a}}) + e_{h_1}(-\dot{\hat{h}}_1).$$

Considering $y$ as a virtual controller, then $y = v_1(x)$

$$\dot{V}_1 = x\left[x\left(1 - \frac{x}{K}\right) - \frac{ax\gamma_1}{1+x} - h_1 x + u_1\right] + e_a(-\dot{\hat{a}}) + e_{h_1}(\dot{\hat{h}}_1). \tag{22}$$

Choosing $v_1 = 0$ and using the controller (15), Ref. (22) becomes

$$\dot{V}_1 = x(-x - xe_{h_1}) + e_a(\dot{\hat{a}}) + e_{h_1}(\dot{\hat{h}}_1). \tag{23}$$

The updated law by the unknown parameter

$$\dot{\hat{a}} = e_a \quad and \quad \dot{\hat{h}}_1 = -x^2 + e_{h_1}. \tag{24}$$

Substituting (24) in (22), we get $\dot{V}_1 = -x^2 - e_a^2 - e_{h_1}^2$ is the negative definite function, where

$$v_1 = 0 \implies \eta_1 = y. \tag{25}$$

Differentiating (25), we get

$$\dot{\eta}_1 = \dot{y} = \frac{axy}{1+x} - byz - h_2 y + u_2. \tag{26}$$

Now, Equation (13) along with Equation (26), we get that

$$\dot{x} = -x - \frac{ax\eta_1}{1+x} - e_{h_1}x. \tag{27}$$

Considering the Lyapunov function of $(x, \eta_1)$,

$$V_2 = V_1 + \frac{1}{2}\eta_1^2 + \frac{1}{2}e_b^2 + \frac{1}{2}e_{h_2}^2. \tag{28}$$

Differentiating (28) with respect to $t$, we get

$$\dot{V}_2 = \quad x\left(-x - \frac{ax\eta_1}{1+x} - e_{h_1}x\right) + \eta_1\left(\frac{ax\eta_1}{1+x} - b\eta_1 z - h_2\eta_1 + u_2\right)$$
$$e_a(-\dot{\hat{a}}) + e_{h_1}(-\dot{\hat{h}}_1) + e_b(-\dot{\hat{b}}) + e_{h_2}(-\dot{\hat{h}}_2). \tag{29}$$

Again, considering a new virtual controller $z = \nu_2(x, y)$ where $\nu_2 = 0$, and using this in (24), we have

$$\dot{V}_2 = -x^2 - e_{h_1}^2 - e_a^2 + e_{h_2}(-\dot{\hat{h}}_2) + \eta_1\left(\frac{-ax^2}{1+x} + \frac{ax\eta_1}{1+x} - h_2\eta_1 + u_2\right). \tag{30}$$

Now, choosing the controller (16) along with (30), we get

$$\dot{V}_2 = -x^2 - e_a^2 - e_{h_1}^2 + e_{h_2}(\dot{\hat{h}}_2) + e_b(-\dot{\hat{b}}) - \eta_1^2. \tag{31}$$

The updated law for the unknown parameter $\dot{\hat{b}}$ and $\dot{\hat{h}}_2$ is

$$\dot{\hat{b}} = e_b \quad and \quad \dot{\hat{h}}_2 = -\eta_1^2 + e_{h_2}. \tag{32}$$

Substituting (32) in (31), we get

$$\dot{V}_2 = -x^2 - e_b^2 - e_a^2 - e_{h_1}^2 - e_{h_2}^2 - \eta_1^2, \tag{33}$$

which is again a negative definite function where

$$\nu_2 = 0 \implies \eta_2 = z. \tag{34}$$

Differentiating (34) with respect to $t$, we have $\dot{\eta}_2 = \dot{z}$.
Now,

$$\dot{\eta}_1 = \frac{ax\eta_1}{1+x} - b\eta_1 z - h_2\eta_1 + u_2, \tag{35}$$

where the controller (17) along with (35) gives

$$\dot{\eta}_1 = -b\eta_1\eta_2 - e_{h_2}\eta_1 + \frac{ax^2}{1+x}, \tag{36}$$

$$\dot{\eta}_2 = fyz - dz + u_3. \tag{37}$$

Now, considering the Lyapunov function of $(x, \eta_1, \eta_2)$,

$$V_3 = V_2 + \frac{1}{2}\eta_2^2 + \frac{1}{2}e_d^2 + \frac{1}{2}e_f^2. \tag{38}$$

Differentiating (38) with respect to $t$ gives

$$
\begin{aligned}
\dot{V}_3 = & -x^2 - e_{h_1}(x^2) - e_{h_2}\eta_1^2 - \eta_1^2 - \eta_1^2 + \eta_2(-b\eta_1^2 + f\eta_1\eta_2 - d\eta_2 + u_3) \\
& + e_a(-\dot{\hat{a}}) + e_b(-\dot{\hat{b}}) + e_f(-\dot{\hat{f}}) + e_{h_1}(-\dot{\hat{h}}_1) + e_{h_2}(\dot{\hat{h}}_2) + e_d(\dot{\hat{d}}).
\end{aligned}
\tag{39}
$$

The unknown parameters $\dot{\hat{a}}, \dot{\hat{b}}, \dot{\hat{f}}, \dot{\hat{h}}_2$ are updated by

$$
\dot{\hat{a}} = e_a, \dot{\hat{b}} = e_b, \dot{\hat{f}} = e_f, \dot{\hat{h}}_2 = -\eta_1^2 + e_{h_2}, (\dot{\hat{d}}) = +d.
\tag{40}
$$

Substituting the updated parameters along with choosing the controller (17) and by updating law (40) in Equation (39), we get $\dot{V}_3 = -x^2 - \eta_1^2 - \eta_2^2 - e_a^2 - e_b^2 - e_f^2 - e_{h_1}^2 - e_{h_2}^2 - e_d^2$ is the negative definite function. Thus, by the Lyapunov stability theorem, systems (10)–(12) is globally asymptotically stable with adaptive back-stepping controllers. □

## 8. Stochastic Analysis

All usual occurrences explicitly in the ecosystem are continuously under random fluctuations of the environment. The stochastic examination of any ecosystem gives an enhanced vision on the dynamic forces of the populace by means of population variances. In a stochastic model, the model parameters oscillate about their average values [24–27]. Therefore, the steady state which we anticipated as permanent will now oscillate around the mean state. The method to measure the mean-square fluctuations of population is proposed by [24] and it was applied by [28] nicely. Furthermore, many researchers like Samanta [29], Maiti, Jana and Samanta [30] have investigated critically the stochastic analysis to interpret local as well as global stability using mean-square fluctuations on population variances.

Now, this segment is meant for the extension of the deterministic model (1), which is formed by adding a noisy term. There are several ways in which environmental noise may be incorporated in the model system (1). External noise may arise from random fluctuations of a finite number of parameters around some known mean values of the population densities around some fixed values. Since the aquatic ecosystem always has unsystematic fluctuations of the environment, it is difficult to define the usual phenomenon as a deterministic ideal. The stochastic investigation enables us to get extra intuition about the continuous changing aspects of any ecological unit. The deterministic model (1) with the effect of random noise of the environment results in a stochastic system (41)–(43) given in the following discussion:

$$
\frac{dx}{dt} = x\left(1 - \frac{x}{K}\right) - \frac{axy}{1+x} - h_1 x + \alpha_1\xi_1(t),
\tag{41}
$$

$$
\frac{dy}{dt} = \frac{axy}{1+x} - byz - h_2 y + \alpha_2\xi_2(t),
\tag{42}
$$

$$
\frac{dz}{dt} = fyz - dz + \alpha_3\xi_3(t),
\tag{43}
$$

where $\alpha_1, \alpha_2, \alpha_3$ are the real constants and $\xi_i(t) = [\xi_1(t), \xi_2(t), \xi_3(t)]$ is a three-dimensional Gaussian white noise process. $E(\xi_i(t)) = 0$, where $i = 1, 2, 3$; $E[\xi_i(t)\xi_j(t)] = \delta_{ij}\delta(t - t')$, where $i = j = 1, 2, 3$; $\delta_{ij}$ is the Kronecker delta function; $\delta$ is the Dirac delta function.

Let

$$
x(t) = u_1(t) + S^*; y(t) = u_2(t) + P^*; z(t) = u_3(t) + T^*,
\tag{44}
$$

$$
\frac{dx}{dt} = \frac{du_1(t)}{dt}; \frac{dy}{dt} = \frac{du_2(t)}{dt}; \frac{dz}{dt} = \frac{du_3(t)}{dt}.
\tag{45}
$$

The linear parts of (41), (42) and (43) are (using (44) and (45))

$$
u_1'(t) = u_1(t)S^* - au_2(t)S^* + \alpha_1\xi_1(t),
\tag{46}
$$

$$u_2'(t) = au_1(t)P^* - bu_3(t)P^* + \alpha_2\xi_2(t), \tag{47}$$

$$u_3'(t) = fu_2(t)T^* + \alpha_3\xi_3(t). \tag{48}$$

Taking the Fourier transform on both sides of (46), (47) and (48), we get

$$i\omega\tilde{u}_1(\omega) = -S^*\tilde{u}_1(\omega) - aS^*\tilde{u}_2(\omega) + \alpha_1\xi_1(\omega), \tag{49}$$

$$i\omega\tilde{u}_2(\omega) = aP^*\tilde{u}_1(\omega) - bP^*\tilde{u}_3(\omega) + \alpha_2\xi_2(\omega), \tag{50}$$

$$i\omega\tilde{u}_3(\omega) = fT^*\tilde{u}_2(\omega) + \alpha_3\xi_3(\omega). \tag{51}$$

The matrix form of (49)–(51) is

$$M(\omega)\tilde{u}(\omega) = \tilde{\xi}(\omega), \tag{52}$$

where

$$M(\omega) = \begin{pmatrix} i\omega + S^* & aS^* & 0 \\ -aP^* & i\omega & bP^* \\ 0 & -fT^* & i\omega \end{pmatrix}; \tilde{u}(\omega) = \begin{bmatrix} \tilde{u}_1(\omega) \\ \tilde{u}_2(\omega) \\ \tilde{u}_3(\omega) \end{bmatrix}; \tilde{\xi}(\omega) = \begin{bmatrix} \eta_1\tilde{\xi}_1(\omega) \\ \eta_2\tilde{\xi}_2(\omega) \\ \eta_3\tilde{\xi}_3(\omega) \end{bmatrix}.$$

Equation (52) can also be written into

$$\tilde{u}(\omega) = [M(\omega)]^{-1}\tilde{\xi}(\omega), \tag{53}$$

where

$$[M(\omega)]^{-1} = \frac{1}{R(\omega) + iI(\omega)} \begin{pmatrix} A_1 & D_1 & G_1 \\ B_1 & E_1 & H_1 \\ C_1 & F_1 & I_1 \end{pmatrix} \tag{54}$$

and

$$A_1 = -\omega^2 + fb_3T^*P^*; \quad C_1 = afP^*T^*; \quad\quad\quad D_1 = i\omega\alpha_1S^*;$$

$$E_1 = -\omega^2 + i\omega S^*; \quad F_1 = i\omega fT^* + fT^*S^*; \quad G_1 = \alpha_1b_3S^*P^*;$$

$$H_1 = -i\omega bP^* - bS^*P^*; \quad I_1 = -\omega^2 + i\omega S^* + a\alpha_1 P^*S^*.$$

Here,

$$|A_1|^2 = X_1^2 + Y_1^2; \quad |B_1|^1 = X_2^2 + Y_2^2; \quad |C_1|^2 = X_3^2 + Y_3^2;$$

$$|D_1|^2 = X_4^2 + Y_4^2; \quad |E_1|^2 = X_5^2 + Y_5^2; \quad |F_1|^2 = X_6^2 + Y_6^2;$$

$$|G_1|^2 = X_7^2 + Y_7^2; \quad |H_1|^2 = X_8^2 + Y_8^2; \quad |I_1|^2 = X_9^2 + Y_9^2,$$

where

$$X_1 = -\omega^2 + fbT^*P^*; Y_1 = 0; X_2 = 0; Y_2 = (a\omega P^*); X_3 = afP^*T^*;$$

$$Y_3 = 0; X_4 = 0; Y_4 = \omega\alpha_1 S^*; X_5 = \omega S^*; X_6 = \omega fT^*; \tag{55}$$

$$Y_6 = fT^*S^*; X_7 = 0; Y_7 = \alpha_1 bS^*P^*; X_8 = bS^*P^*; Y_8 = \omega bP^*;$$

$$X_9 = -\omega^2 + a\alpha_1 S^*P^*; Y_9 = \omega S^*.$$

$|M(\omega)|^2 = [R(\omega)]^2 + [I(\omega)]^2$, where $R(\omega) = b_3fT^*P^*S^* - S^*\omega^2$ and $I(\omega) = \omega^2 + \omega b_3fT^*P^* + \alpha_1 a\omega S^*P^*$.

If the function $Y(t)$ has a zero mean value, then the fluctuation intensity (variance) of its components in the frequency interval $[\omega, \omega + d\omega]$ is $S_Y(\omega)d\omega$. $S_Y(\omega)$ is the spectral density of $Y$ and is defined as

$$S_Y(\omega) = \lim_{\tilde{T}\to\infty} \frac{|\tilde{Y}(\omega)|^2}{\tilde{T}}. \tag{56}$$

If $Y$ has a zero mean value, the inverse transform of $S_Y(\omega)$ is the auto covariance function

$$C_Y(\tau) = \frac{1}{2\pi} \int\limits_{-\infty}^{\infty} S_Y(\omega) e^{i\omega\tau} d\omega. \tag{57}$$

The corresponding variance of fluctuations in $Y(t)$ is given by

$$\sigma_Y^2 = C_Y(0) = \frac{1}{2\pi} \int\limits_{-\infty}^{\infty} S_Y(\omega) d\omega \tag{58}$$

and the auto-correlation function is the normalized auto-covariance

$$P_Y(\tau) = \frac{C_Y(\tau)}{C_Y(0)}. \tag{59}$$

For a Gaussian white noise process, it is

$$
\begin{aligned}
S_{\xi_i\xi_j}(\omega) &= \lim_{\tilde{T}\to+\infty} \frac{E\left[\tilde{\xi}_i(\omega\tilde{\xi}_j(\omega))\right]}{\tilde{T}} \\
&= \lim_{\hat{T}\to+\infty} \frac{1}{\hat{T}} \int\limits_{\frac{-\tilde{T}}{2}}^{\frac{\tilde{T}}{2}} \int\limits_{\frac{-\tilde{T}}{2}}^{\frac{\tilde{T}}{2}} \left[\tilde{\xi}_i(t)\tilde{\xi}_j(t')\right] e^{-i\omega(t-t')} dt\, dt' \\
&= \delta_{ij}.
\end{aligned} \tag{60}
$$

From (54), we have

$$\tilde{u}_i(\omega) = \sum_{j=1}^{3} K_{ij}(\omega)\tilde{\xi}_j(\omega); \quad i = 1, 2, 3. \tag{61}$$

From (59), we have

$$S_{ui}(\omega) = \sum_{j=1}^{3} \eta_j |K_{ij}(\omega)|^2; \quad i = 1, 2, 3, \tag{62}$$

where

$$K_{ij}(\omega) = [M(\omega)]^{-1}.$$

Hence, by (61) and (62), the intensities of fluctuations in the variable $u_i$ ($i = 1, 2, 3$) are given by

$$\sigma_{u_i}^2 = \frac{1}{2\pi} \sum_{j=1}^{3} \int\limits_{-\infty}^{\infty} \eta_j |K_{ij}(\omega)|^2 d\omega; \quad i = 1, 2, 3 \tag{63}$$

and by (54), we obtain

$$\sigma_{\mu_1}^2 = \frac{1}{2\pi} \left\{ \int\limits_{-\infty}^{\infty} \eta_1 \left|\frac{Adj(1)}{|M(\omega)|}\right|^2 d\omega + \int\limits_{-\infty}^{\infty} \eta_2 \left|\frac{Adj(2)}{|M(\omega)|}\right|^2 d\omega + \int\limits_{-\infty}^{\infty} \eta_3 \left|\frac{Adj(3)}{|M(\omega)|}\right|^2 d\omega \right\}, \tag{64}$$

$$\sigma_{\mu_2}^2 = \frac{1}{2\pi} \left\{ \int\limits_{-\infty}^{\infty} \eta_1 \left|\frac{Adj(4)}{|M(\omega)|}\right|^2 d\omega + \int\limits_{-\infty}^{\infty} \eta_2 \left|\frac{Adj(5)}{|M(\omega)|}\right|^2 d\omega + \int\limits_{-\infty}^{\infty} \eta_3 \left|\frac{Adj(6)}{|M(\omega)|}\right|^2 d\omega \right\}, \tag{65}$$

$$\sigma_{\mu_1}^2 = \frac{1}{2\pi} \left\{ \int\limits_{-\infty}^{\infty} \eta_1 \left|\frac{Adj(7)}{|M(\omega)|}\right|^2 d\omega + \int\limits_{-\infty}^{\infty} \eta_2 \left|\frac{Adj(8)}{|M(\omega)|}\right|^2 d\omega + \int\limits_{-\infty}^{\infty} \eta_3 \left|\frac{Adj(9)}{|M(\omega)|}\right|^2 d\omega \right\}. \tag{66}$$

From (55), (64), (65) and (66),

$$\sigma_{u_1}^2 = \frac{1}{2\pi} \left\{ \int_{-\infty}^{\infty} \frac{1}{R^2(\omega) + I^2(\omega)} \left[ \alpha_1(X_1^2 + Y_1^2) + \alpha_2(X_2^2 + Y_2^2) + \alpha_3(X_3^2 + Y_3^2) \right] d\omega \right\}, \tag{67}$$

$$\sigma_{u_2}^2 = \frac{1}{2\pi} \left\{ \int_{-\infty}^{\infty} \frac{1}{R^2(\omega) + I^2(\omega)} \left[ \alpha_1(X_4^2 + Y_4^2) + \alpha_2(X_5^2 + Y_5^2) + \alpha_3(X_6^2 + Y_6^2) \right] d\omega \right\}, \tag{68}$$

$$\sigma_{u_3}^2 = \frac{1}{2\pi} \left\{ \int_{-\infty}^{\infty} \frac{1}{R^2(\omega) + I^2(\omega)} \left[ \alpha_1(X_7^2 + Y_7^2) + \alpha_2(X_8^2 + Y_8^2) + \alpha_3(X_9^2 + Y_9^2) \right] d\omega \right\}, \tag{69}$$

where

$$|M(\omega)| = R(\omega) + iI(\omega).$$

If we are interested in the dynamics of stochastic process (41)–(69) with either $\alpha_1 = 0$ or $\alpha_2 = 0$ or $\alpha_3 = 0$, the population variances are

if $\alpha_1 = 0, \alpha_2 = 0$, then $\sigma_{u1}^2 = \frac{\alpha_3}{2\pi} \int_{-\infty}^{\infty} \frac{(X_3^2 + Y_3^2)}{R^2(\omega) + I^2(\omega)} d\omega; \sigma_{u2}^2 = \frac{\alpha_3}{2\pi} \int_{-\infty}^{\infty} \frac{(X_6^2 + Y_6^2)}{R^2(\omega) + I^2(\omega)} d\omega;$

$\sigma_{u3}^2 = \frac{\alpha_3}{2\pi} \int_{-\infty}^{\infty} \frac{(X_9^2 + Y_9^2)}{R^2(\omega) + I^2(\omega)} d\omega.$

If $\alpha_2 = 0, \alpha_3 = 0$, then $\sigma_{u1}^2 = \frac{\alpha_1}{2\pi} \int_{-\infty}^{\infty} \frac{(X_1^2 + Y_1^2)}{R^2(\omega) + I^2(\omega)} d\omega; \sigma_{u2}^2 = \frac{\alpha_1}{2\pi} \int_{-\infty}^{\infty} \frac{(X_4^2 + Y_4^2)}{R^2(\omega) + I^2(\omega)} d\omega;$

$\sigma_{u3}^2 = \frac{\alpha_1}{2\pi} \int_{-\infty}^{\infty} \frac{(X_7^2 + Y_7^2)}{R^2(\omega) + I^2(\omega)} d\omega.$

If $\alpha_3 = 0, \alpha_1 = 0$, then $\sigma_{u1}^2 = \frac{\alpha_2}{2\pi} \int_{-\infty}^{\infty} \frac{(X_2^2 + Y_2^2)}{R^2(\omega) + I^2(\omega)} d\omega; \sigma_{u2}^2 = \frac{\alpha_2}{2\pi} \int_{-\infty}^{\infty} \frac{(X_5^2 + Y_5^2)}{R^2(\omega) + I^2(\omega)} d\omega;$

$\sigma_{u3}^2 = \frac{\alpha_2}{2\pi} \int_{-\infty}^{\infty} \frac{(X_8^2 + Y_8^2)}{R^2(\omega) + I^2(\omega)} d\omega.$

Equations (67)–(69) give three variations of the inhabitants. The integrations over the real line can be estimated, which gives the variations of the inhabitants.

## 9. Mathematical Model with Delay

In this section, we establish some conditions for oscillations of all positive solutions of the delay system

$$\frac{dx}{dt}(t) = x(t) \left( 1 - h_1 - \frac{x(t-\tau)}{K} - \frac{ay(t-\tau)}{1 + x(t-\tau)} \right), \tag{70}$$

$$\frac{dy}{dt}(t) = y(t) \left( -h_2 + \frac{ax(t-\tau)}{1 + x(t-\tau)} - bz(t-\tau) \right), \tag{71}$$

$$\frac{dz}{dt}(t) = z(t) \left( -d + fy(t-\tau) \right). \tag{72}$$

Here, the parameter $\tau \in \mathbb{R}^+$ is the delay. This proposed system is concerned not only with the present number of predator and prey but also with the number of predator and prey in the past. If $t$ is the present time, then $(t\text{-}\tau)$ is the past.

According to Krisztin [31], a solution of (70)–(72), $(x(t), y(t), z(t))$ is called *oscillatory* if every component has arbitrary large zeros; otherwise, $(x(t), y(t), z(t))$ is said to be a *non-oscillatory* solution. Whenever all solutions of (70)–(72) are oscillatory, we will say that (70)–(72) is an *oscillatory* system.

In [32], Kubiaczyk and Saker studied the oscillatory behavior of the delay differential equation

$$x'(t) + px(t) - \frac{qx(t)}{r + x^n(t - \tau)} = 0,$$

where $p$, $q$, $r$, $\tau \in \mathbb{R}^+$. Using similar methods to liberalize each equation of the delay system, we will establish conditions for oscillations of all positive solutions of the system.

Now, we will analyze the system of (70)–(72) with the following initial conditions:

$$x(\theta) \geq 0, \; y(\theta) \geq 0, \; z(\theta) \geq 0, \; \theta \in (-\tau, 0), \; (\tau \neq 0). \tag{73}$$

Using the same arguments that we got in Theorem 1, we can establish the following theorem:

**Theorem 7.** *All solutions of $(x(t), y(t), z(t))$ of systems (70)–(72) with the initial condition (73) are positive for all $t \geq 0$.*

Easily, we can see that the equilibrium point remains the same when we have the delay system. However, it is important to know the oscillatory behavior of the solutions around the equilibrium points.

**Theorem 8.** *If there exist a $\lambda \in \mathbb{R}$ such that*

$$\min_{\lambda \in \mathbb{R}} \left( \lambda^3 - \alpha\lambda^2 e^{-\lambda\tau} - (d\sigma + \beta\gamma) \lambda e^{-2\lambda\tau} + d\sigma\alpha e^{-3\lambda\tau} \right) > 0, \tag{74}$$

*where $\alpha = -\frac{x^*}{k}$, $\beta = -\frac{ad}{f(1+x^*)}$, $\gamma = \frac{ax^*}{1+x^*}$ and $\sigma = -dz^*$; then, all solutions of the system (70)–(72) oscillate around $E_3$.*

**Proof.** Let us consider the system (70)–(72). Let

$$x(t) = x^* e^{\phi_1(t)}, \tag{75}$$

$$y(t) = y^* e^{\phi_2(t)}, \tag{76}$$

$$z(t) = z^* e^{\phi_3(t)}. \tag{77}$$

Then, $(x(t), y(t), z(t))$ oscillate around $(x^*, y^*, z^*)$ if $(\phi_1(t), \phi_2(t), \phi_3(t))$ oscillate around $(0, 0, 0)$. From (70)–(72) and (75)–(77), we have

$$\begin{aligned}
\frac{d\phi_1}{dt}(t) &= 1 - h_1 - \frac{x^* e^{\phi_1(t-\tau)}}{K} - \frac{ay^* e^{\phi_2(t-\tau)}}{1 + x^* e^{\phi_1(t-\tau)}} \\
&= 1 - h_1 - \frac{x^*}{K} - \frac{ay^*}{1 + x^* e^{\phi_1(t-\tau)}} - \frac{x^*}{K} \left( e^{\phi_1(t-\tau)} - 1 \right) \\
&\quad - \frac{ay^*}{1 + x^* e^{\phi_1(t-\tau)}} \left( e^{\phi_2(t-\tau)} - 1 \right),
\end{aligned}$$

$$\begin{aligned}
\frac{d\phi_2}{dt}(t) &= -h_2 + \frac{ax^* e^{\phi_1(t-\tau)}}{1 + x^* e^{\phi_1(t-\tau)}} - bz^* e^{\phi_3(t-\tau)} \\
&= -h_2 + \frac{ax^*}{1 + x^* e^{\phi_1(t-\tau)}} - bz^* + \frac{ax^*}{1 + x^* e^{\phi_1(t-\tau)}} \left( e^{\phi_1(t-\tau)} - 1 \right) \\
&\quad - bz^* \left( e^{\phi_3(t-\tau)} - 1 \right),
\end{aligned}$$

$$\frac{d\phi_3}{dt}(t) = -d + fy^* + fy^* \left( e^{\phi_2(t-\tau)} - 1 \right).$$

Let

$$f(u) = e^u - 1.$$

Note that

$$uf(u) > 0 \text{ for } u > 0, \text{ and } \lim_{u \to 0} \frac{f(u)}{u} = 1. \tag{78}$$

Moreover, since $(x^*, y^*, z^*)$ is the equilibrium point $E_3$, we have

$$1 - h_1 - \frac{x^*}{K} - \frac{ay^*}{1+x^*} = 0, \quad -h_2 + \frac{ax^*}{1+x^*} - bz^* = 0, \text{ and } -d + fy^* = 0.$$

Thus, the linearized system associated with the system (75)–(77) is given by

$$\frac{dm_1}{dt}(t) = -\frac{x^*}{K} m_1(t - \tau) - \frac{ay^*}{1+x^*} m_2(t - \tau), \tag{79}$$

$$\frac{dm_2}{dt}(t) = \frac{ax^*}{1+x^*} m_1(t - \tau) - bz^* m_3(t - \tau), \tag{80}$$

$$\frac{dm_3}{dt}(t) = fy^* m_2(t - \tau), \tag{81}$$

and every solution of (79)–(81) oscillates if and only if the characteristic equation has no real roots (see Theorem 5.1.1 in [21]), i.e.,

$$\det[\lambda I - A] \neq 0 \tag{82}$$

for all $\lambda \in \mathbb{R}$. Equation (82) is equivalent to the equation

$$\lambda^3 + \alpha \lambda^2 e^{-\lambda\tau} - (d\sigma + \beta\gamma) \lambda e^{-2\lambda\tau} + d\sigma\alpha e^{-3\lambda\tau} = 0,$$

where $\alpha = \frac{x^*}{k}$, $\beta = -\frac{ad}{f(1+x^*)}$, $\gamma = \frac{ax^*}{1+x^*}$ and $\sigma = dz^*$. In fact,

$$\lim_{\lambda \to +\infty} \left( \lambda^3 + \alpha \lambda^2 e^{-\lambda\tau} - (d\sigma + \beta\gamma) \lambda e^{-2\lambda\tau} + d\sigma\alpha e^{-3\lambda\tau} \right) = +\infty$$

and

$$\lim_{\lambda \to -\infty} \left( \lambda^3 + \alpha \lambda^2 e^{-\lambda\tau} - (d\sigma + \beta\gamma) \lambda e^{-2\lambda\tau} + d\sigma\alpha e^{-3\lambda\tau} \right) = +\infty, \text{ since } d\sigma\alpha > 0.$$

Then, by (74) and (78), systems (79)–(81) will start oscillating and then all the solutions of systems (75)–(77) will also oscillate. $\square$

**Example 1.** *Let the parameters of systems (70)–(72) be $a = 1.5$, $d = 0.2$, $b = 0.3$, $f = 0.65$, $h_1 = 0.5$, $h_2 = 0.3$, $K = 9$ and $\tau = 2$. In this case, condition (74) becomes*

$$\min_{\lambda \in \mathbb{R}} \left( \lambda^3 + 0.3995837778 \lambda^2 e^{-\lambda\tau} + 0.234339529 \lambda e^{-2\lambda\tau} + 0.046546037 e^{-3\lambda\tau} \right) > 0.045$$

*and consequently all solutions of the system oscillate around the equilibrium point* $(3.596254004, 0.307692308, 2.912157597)$.

## 10. Numerical Simulations

Analytical studies can never complete without numerical verification of the results. Moreover, it may be noted that, as the parameters of the model are not based on the real world observation, the main features described by the simulations presented in this section should be considered from a qualitative rather than a quantitative point of view. We choose the parameters of system (1) as

$a = 1.5, b = 0.3, f = 0.65, h_1 = 0.5, h_2 = 0.3, d = 0.2, K = 8$ with the initial densities $x(0) = 2$, $y(0) = 1.8, z(0) = 1$ and observe the dynamical behaviour of system (1). Figure 1a shows that the equilibrium point $E_1$ is locally asymptotically stable and the corresponding phase-portrait is shown in Figure 1b. Figure 2a shows that the equilibrium point $E_2$ is locally asymptotically stable and the corresponding phase-portrait is shown in Figure 2b. Figure 3a shows that the co-existence equilibrium point $E_3$ is locally asymptotically stable and the corresponding phase portrait is shown in Figure 3b. From Figure 4a–d, if all other parameters are fixed and varying transmission rate $a = 1.5$ to $a = 2$, we observe that oscillation settles down into a stable situation for all three of the species. Stability around this state implies extinction of the infected prey. This study interestingly suggests that the harvesting of both prey prevent limit cycle oscillations and the combined effect of both harvests also prevent disease propagation in the system. We also conclude that the inclusion of stochastic perturbation creates a significant change in the intensity of populations because change of responsive noise parameters causes chaotic dynamics with low, medium and high variances of oscillations (see Figures 5–7).

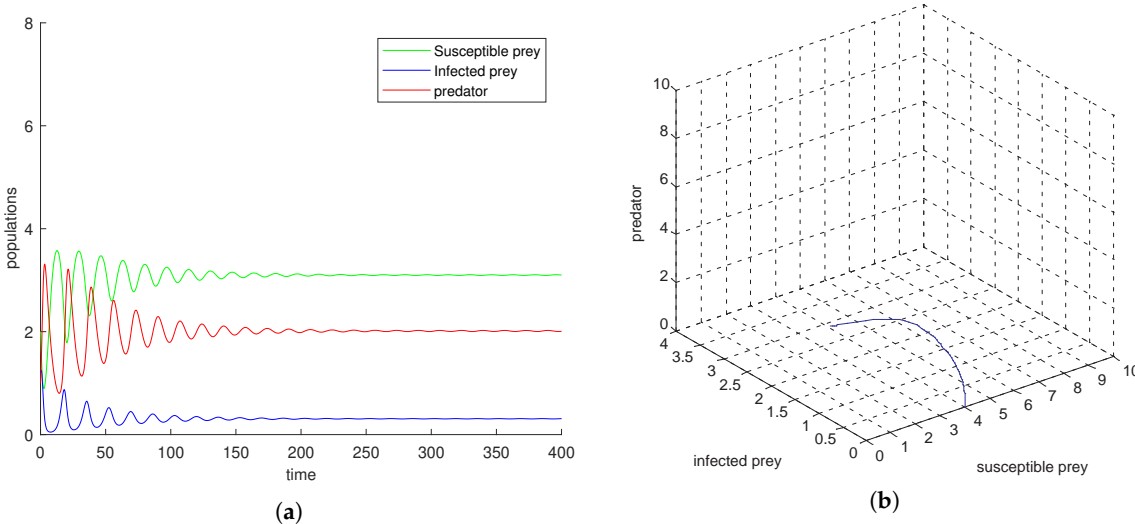

(**a**)

(**b**)

**Figure 1.** (**a**) time series evolution of the populations of the system (1); (**b**) phase-space trajectories corresponding to the stabilities of the population with the parameters $a = 1.5, b = 0.3, f = 0.65$, $h_1 = 0.5, h_2 = 0.3, d = 0.2, K = 8$.

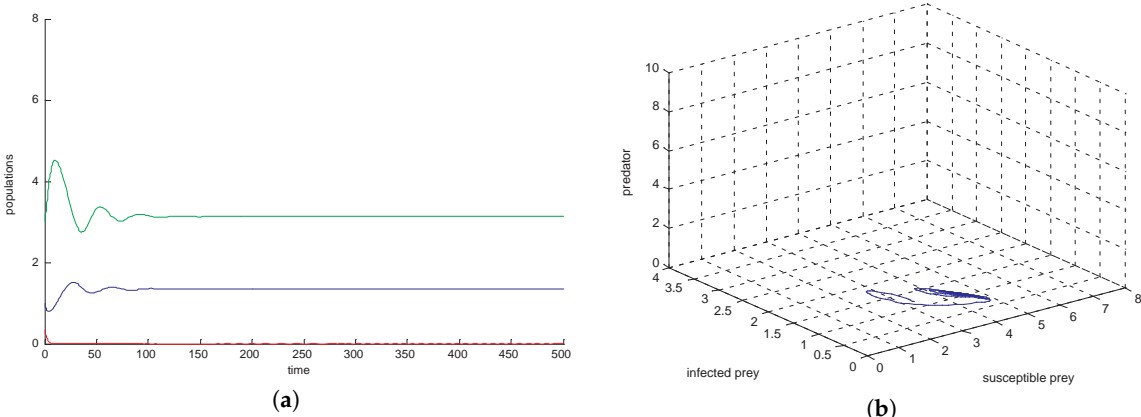

(**a**)

(**b**)

**Figure 2.** (**a**) time series evolution of the populations of system (1); (**b**) phase-space trajectories corresponding to the stabilities of the population with the parameters $a = 1.5, b = 0.3, f = 0.65$, $h_1 = 0.5, h_2 = 0.3, d = 0.2, K = 9$.

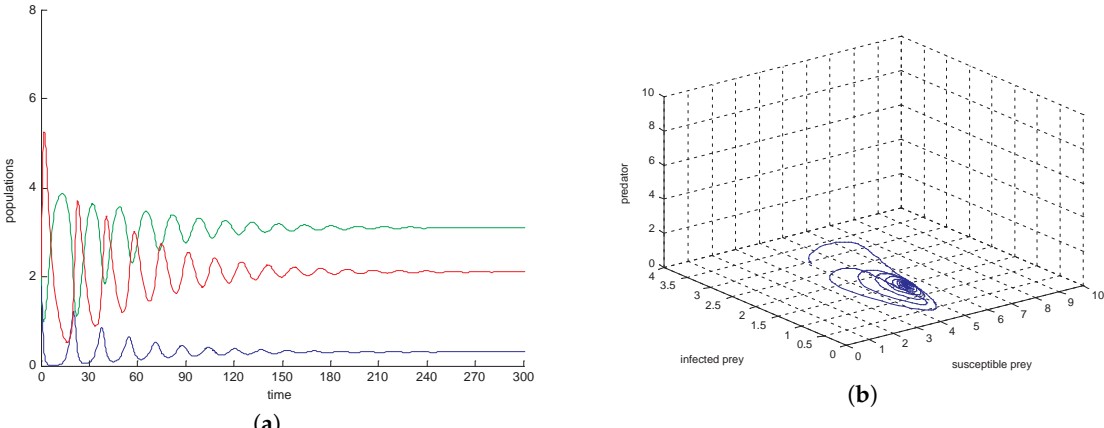

**Figure 3.** (**a**) time series evolution of the populations of system (1); (**b**) phase-space trajectories corresponding to the stabilities of the population with the parameters $a = 1.65$, $b = 0.3$, $f = 0.65$, $h_1 = 0.5$, $h_2 = 0.3$, $d = 0.2$, $K = 9$.

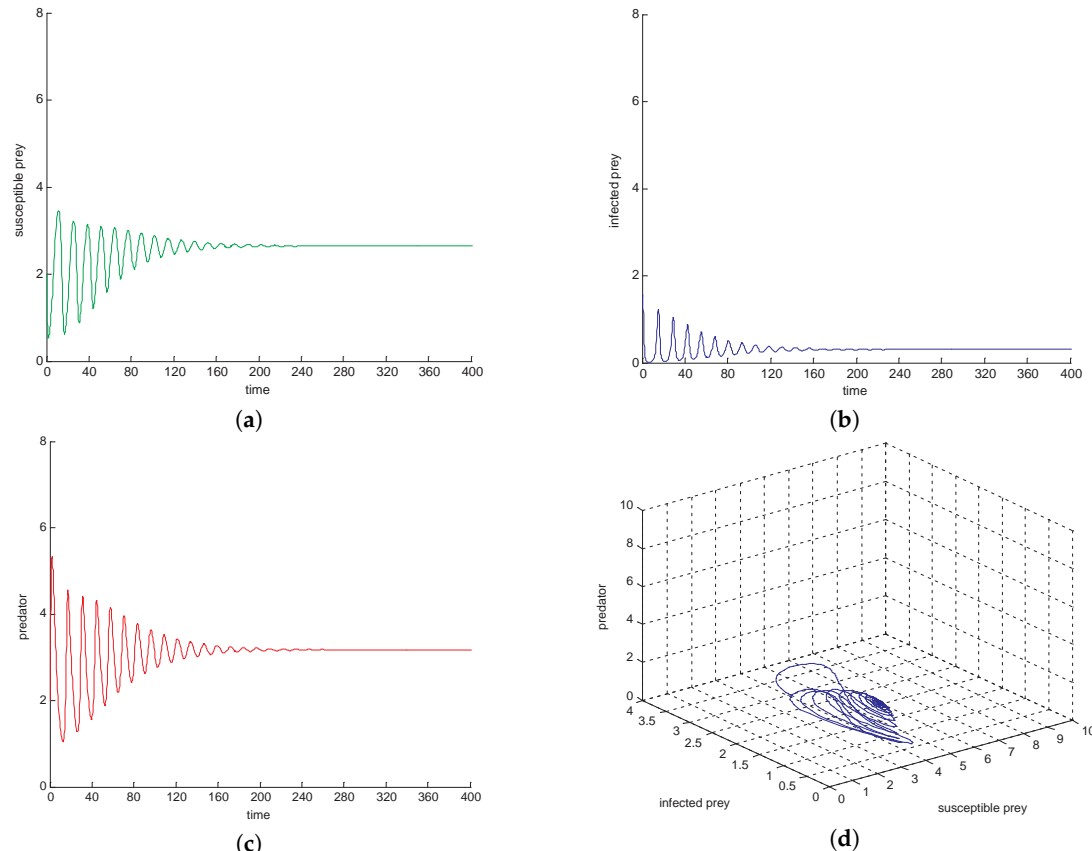

**Figure 4.** (**a**–**c**) time series evolution of the populations of system (1); (**d**) phase-space trajectories corresponding to the stabilities of the population with the parameters $a = 2$, $b = 0.3$, $f = 0.65$, $h_1 = 0.5$, $h_2 = 0.3$, $d = 0.2$, $K = 9$.

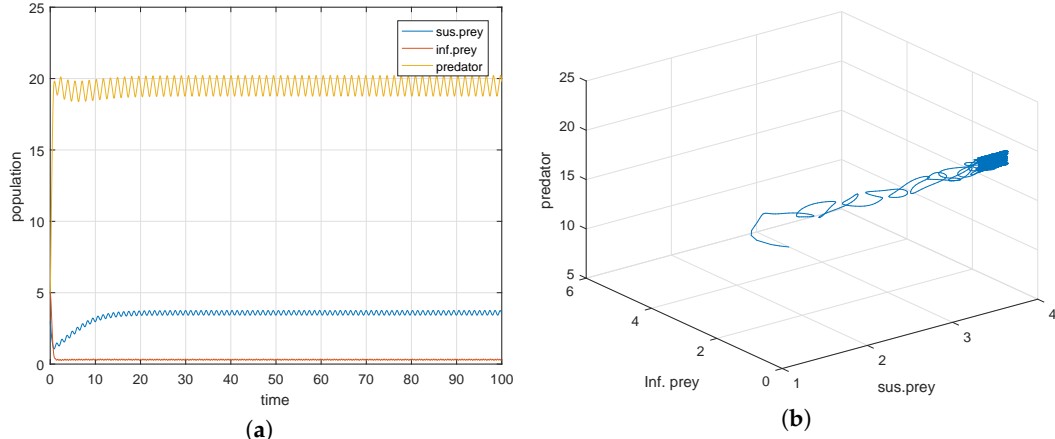

**Figure 5.** (**a**) the oscillations of populations against time with low intensities (low noise) of parameters $\alpha_1 = 1$, $\alpha_2 = 2$, $\alpha_3 = 3$; (**b**) the phase-portrait diagram of populations under random low noise of parameters $\alpha_1 = 1$, $\alpha_2 = 2$, $\alpha_3 = 3$.

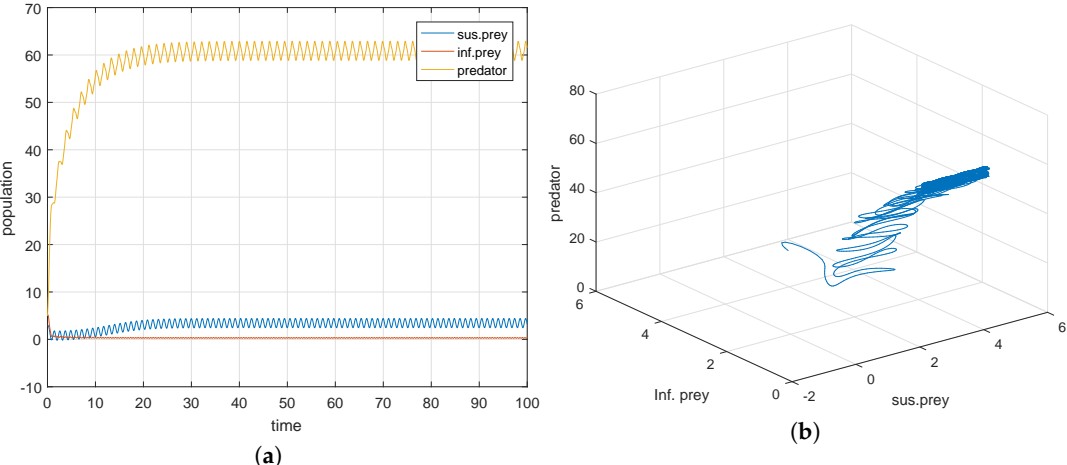

**Figure 6.** (**a**) the oscillations of populations against time with medium intensities (medium noise) of parameters $\alpha_1 = 6$, $\alpha_2 = 8$, $\alpha_3 = 8$; (**b**) the phase-portrait diagram of populations under random medium noise of parameters $\alpha_1 = 6$, $\alpha_2 = 8$, $\alpha_3 = 8$.

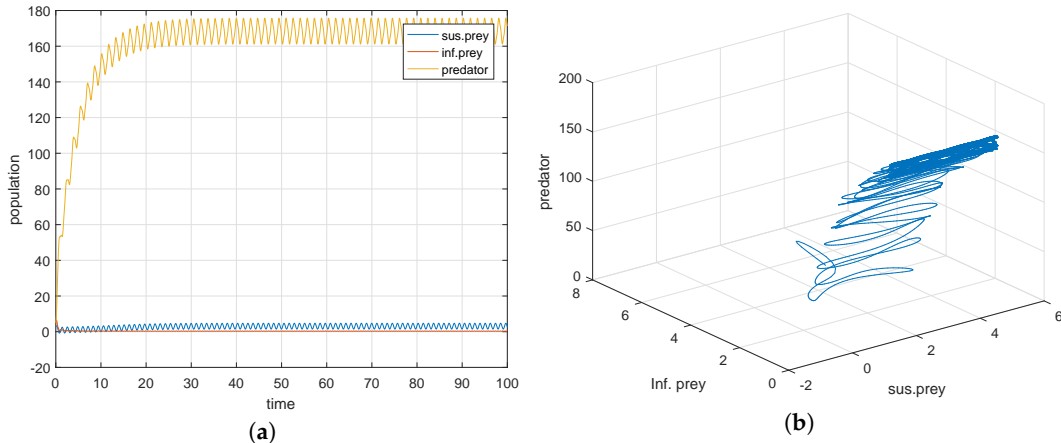

**Figure 7.** (**a**) the oscillations of populations against time with high intensities (high noise) of parameters $\alpha_1 = 10$, $\alpha_2 = 20$, $\alpha_3 = 30$; (**b**) the phase-portrait diagram of populations under random high noise of parameters $\alpha_1 = 10$, $\alpha_2 = 20$, $\alpha_3 = 30$.

## 11. Conclusions

In this paper, we have studied stability of a diseased model of susceptible, infected prey and predators around an interior steady state. The positivity of the solutions and boundedness of the system together with stability analysis of boundary equilibrium providing all the necessary information to establish persistence of the system. The deterministic situation theoretical epidemiologists are usually guided by an implicit assumption that most epidemic models (we observe in nature) correspond to stable equilibrium of models. In Theorem 3, we gave the condition for stable co-existence. In Theorem 4, we proved the global stability by using a Lyapunov function. We also worked out the condition for which all three species will persist. The controllability conditions and the conditions for global asymptotic stability have been obtained by using the adaptive back-stepping control approach by using a suitable Lyapunov function. We also studied the stochastic perturbation of model (1), which generates a significant change in the intensity of populations due to low, medium and high variances of oscillations.

**Author Contributions:** Kalyan Das designed the mathematical model and did the entire analysis with compilation; M. N. Srinivas focused on the stochastic analysis part; V. Madhusudanan contributed to the section of adaptive back-stepping control analysis; Sandra Pinelas performed the analysis of the model with discrete time delay; the numerical simulations were carried out jointly by Kalyan Das and M. N. Srinivas.

**Acknowledgments:** The authors are grateful to the anonymous referee for their helpful comments and valuable suggestions, which led to the improvement of the manuscript.

**Conflicts of Interest:** The authors declare no conflict of interest.

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
