# Peer review of "Mathematical Analysis of a Prey–Predator System: An Adaptive Back-Stepping Control and Stochastic Approach"

_mca, doi:10.3390/mca24010022_

Round 1

Reviewer 1 Report

In this manuscript, the author provide a mathematical model for prey predator system with susceptible, infected prey and a predator. The authors investigate positivity, boundedness, local and global stability for the model. The uniformly persistent of the system is proved by the method of Lyapunov average function.  The system is studied with adaptive back step control and it is proved that the system is stable about its steady state with nonlinear feedback control. The authors also conclude that the inclusion of stochastic perturbation creates a significant change in the intensity of populations due to change of responsive parameters causes chaotic dynamics with low, medium and high variances of oscillations. Finally, the authors provide some numerical simulations to validate the theoretical findings.

The authors should consider the following comments:

1)    The main contribution is not clear.

2)    In order to improve the overall presentation of the work, please  check the English grammar 3) Clarify the biological reason for incorporating time lag in the system 

9)    In introduction section last paragraph; the description of section 9 is not matching with the title of the section.

10)    The way of labeling the system is confusing i.e. system (1)-(3). It is better to write it as system (1)….

11)    The reference in Theorem 2 is not appear Birkoff[?].

12)    Lines 142,143,144 and 145, are repeated the same as in section 2.

13)     Numerical simulation section needs  improvement, in this section the authors discussed figures 1-4 only, the captions of the figures are not informative.

Reviewer 2 Report

This paper discusses a diseased prey-predator system involving adaptive back step control. The system is investigated for its dynamical behaviours such as boundedness, local stability analysis. The global stability of the system is derived using Lyapunov function. The uniform persistence condition for the system is obtained. The proposed system is studied with adaptive back step control and it is proved that the system stabilizes about its steady state with nonlinear feedback control. The value of the system is described signi cantly by the environmental stochasticity in the form of Gaussian white noise.
The model considered in the paper is well motivated, interesting and needs investigations. Mathematical results are correct and interesting from application point of view. The proofs are rather standard and technical. Therefore, I am willing to recommend this paper for publication in MCA.

Author Response

The paper has been recommend  for publication in MCA by the Reviewer's 2. 

Round 2

Reviewer 3 Report

The authors have addressed all the comments and made corresponding changes. 

Now, it is a well written and well organized manuscript. 

Therefore, I recommend this paper for publication in the journal.